# Rates and drivers of aboveground carbon accumulation in global monoculture plantation forests

Jacob J. Bukoski [1,2 ✉], Susan C. Cook-Patton [3,4], Cyril Melikov[2,5], Hongyi Ban [2], Jessica L. Chen [2], Elizabeth D. Goldman[6], Nancy L. Harris [6] & Matthew D. Potts [2,7]

Restoring forest cover is a key action for mitigating climate change. Although monoculture plantations dominate existing commitments to restore forest cover, we lack a synthetic view of how carbon accumulates in these systems. Here, we assemble a global database of 4756 field-plot measurements from monoculture plantations across all forested continents. With these data, we model carbon accumulation in aboveground live tree biomass and examine the biological, environmental, and human drivers that influence this growth. Our results identify four-fold variation in carbon accumulation rates across tree genera, plant functional types, and biomes, as well as the key mediators (e.g., genus of tree, endemism of species, prior land use) of variation in these rates. Our nonlinear growth models advance our understanding of carbon accumulation in forests relative to mean annual rates, particularly during the next few decades that are critical for mitigating climate change.

[1] Conservation International, Arlington, VA, USA. [2] Department of Environmental Science, Policy, and Management, University of California, Berkeley, CA, USA. [3] The Nature Conservancy, Arlington, VA, USA. [4] Smithsonian Conservation Biology Institute, Front Royal, VA, USA. [5] Environmental Defense Fund, New York, NY, USA. [6] World Resources Institute, Washington, DC, USA. [7] Carbon Direct, Inc., New York, NY, USA. ✉email: jbukoski@berkeley.edu

All pathways to limiting global warming to 1.5 °C by 2100 involve substantial removals of atmospheric $CO_2$, highlighting the importance of enhancing carbon capture and storage by forests[1]. Although a range of options exist for successfully restoring forest cover to landscapes, actors across the public, private and non-profit sectors have commonly interpreted this as a need to plant trees[2,3]. For example, two thirds of high-level commitments for tropical forest restoration involve planting and almost half involve the establishment of monoculture plantations[4]. Despite plantations' dominance as a forest restoration strategy, we lack a robust understanding of how much carbon can be captured within monoculture plantations.

Tree planting is controversial because it can negatively impact ecosystems when poorly implemented[5]. In native grasslands, for example, planting trees can reduce endemic biodiversity richness and negatively impact grassland ecosystem functioning[6]. Elsewhere, plantations—especially exotic monocultures—may have limited or adverse biodiversity impacts depending on the species planted and the prior land cover type[7,8]. However, tree planting can be an effective action to mitigate climate change by sequestering atmospheric carbon when done appropriately[9] and monoculture plantations may also be the most viable option to restore forest cover in areas where economic returns are paramount. Further, we need rapid and near-term removals of atmospheric carbon to minimize near-term climate change and plantation forests may sequester carbon slightly more rapidly than naturally regenerating forests, particularly during early phases of establishment[1,10].

Plantation forestry is a well-established practice for growing trees that has existed for centuries[5]. Plantation forest managers have adopted practices from the agricultural sector as well as developed silvicultural methods to improve the growth, form, and yields of trees[11–13]. As such, plantation managers face myriad decisions in the establishment and management of plantations[14]. Not only are managers tasked with locating plantation forests, but they must also make decisions on what species to plant as well as how to manage the trees over short and long-term timeframes. At local scales with relatively constrained conditions (e.g., *Pinus taeda* plantations in the southern United States), these decisions

are well understood. However, there is widespread desire to invest internationally in forests for their climate benefits and the consequences of management decisions on climate outcomes at global scales are poorly understood[15]. Specifically, we lack even a systematic understanding of the magnitude and rate of carbon accumulation in monoculture plantations, and how that varies by drivers such as species, location, or management type.

Here, we systematically reviewed the literature and extracted, from 424 publications, empirical measurements of carbon in aboveground biomass of monoculture plantations. We then analyzed a suite of potential biological, environmental, and human drivers that may explain variation in aboveground carbon accumulation, parameterized growth functions to predict carbon stocks as a function of time, and derived default carbon accumulation rates. Assessing the full climate mitigation potential of plantations requires accounting for the fate of carbon stored in biomass. Although many plantations produce long-lived harvested wood products that can stock large quantities of carbon in the built environment, others produce short-lived harvested wood products (e.g., paper) with limited potential for climate change mitigation. Accurately accounting for the climate impacts of harvested wood products requires context-dependent information that is not yet available for global studies such as ours. Our objective was therefore to improve understanding of a fundamental building block—the rates and drivers of carbon accumulation in monoculture plantation forests—for assessing the climate mitigation potential of these systems. In doing so, our findings facilitate improved understanding of the important and controversial[4] role that monoculture plantations, a dominant global reforestation strategy, may play in mitigating climate change.

## Results & discussion

### Database representativeness of global monoculture plantations.
Our database includes 4756 measurements of carbon in aboveground live tree biomass in monoculture plantations, collected from 829 distinct sites across the globe (Fig. 1). These sites were primarily in Asia (59%), Europe (16%), North America (15%), and South America (4%), with the remainder located in

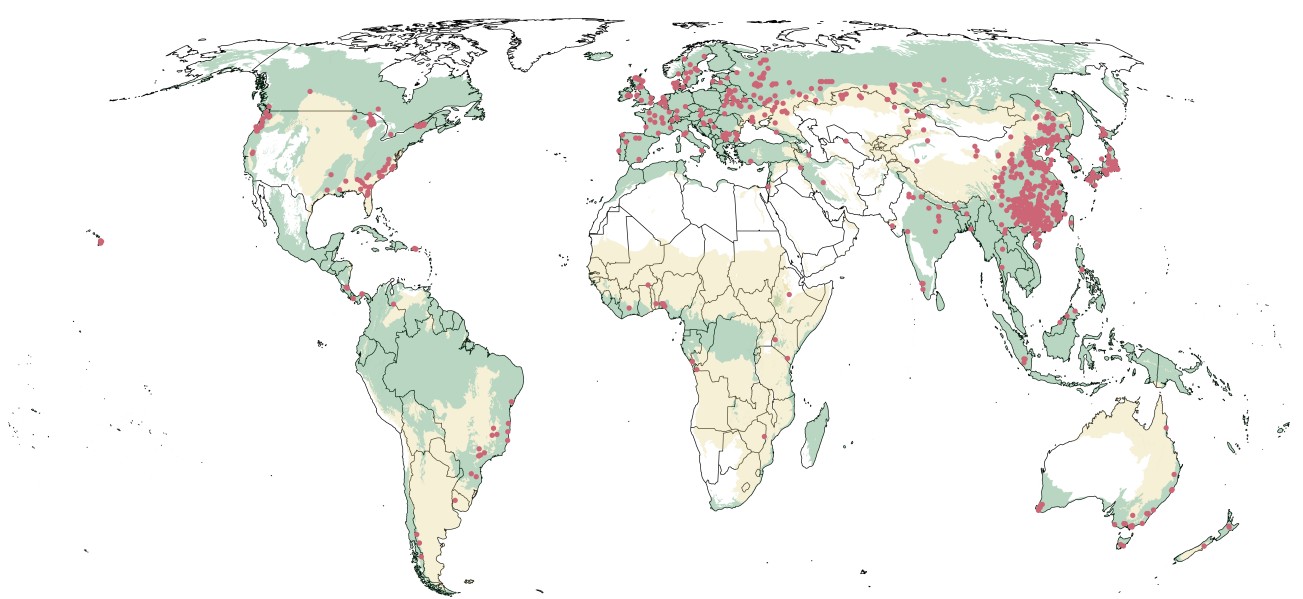

**Fig. 1 Distribution of sites within the database.** We identified a total of 4756 observations of aboveground biomass in plantations, spread across 829 sites. Forested biomes are displayed in light green whereas grassland, savannas, and shrubland biomes are displayed in pale yellow. We distinguish forested and savanna biomes separately since trees in savannas can have strongly negative consequences.

**Table 1 Results of linear mixed effects regression analysis of potential drivers on aboveground carbon accumulation in plantations.**

| Variable | Model Form | No. Obs. | Significance of Effect (*F*-values) | |
|---|---|---|---|---|
| | | | **Driver** | **Interaction** |
| Genus | AGC ~ Stand Age [a] Genus | 4349 | 13.7[c] | 18.4[c] |
| Endemism | AGC ~ Stand Age [a] Endemism | 4705 | 0.3 | 52.8[c] |
| *Plant Traits* | | | | |
| Leaf Type | AGC ~ Stand Age [a] Leaf Type | 4753 | 17.2[c] | 0.8 |
| Leaf Compoundness | AGC ~ Stand Age [a] Leaf Compoundness | 4753 | 3.5 | 2.7 |
| Leaf Phenology | AGC ~ Stand Age [a] Leaf Phenology | 4753 | 2.2 | 0.6 |
| Nitrogen Fixation | AGC ~ Stand Age [a] Nitrogen Fixation | 4753 | 9.1[b] | 0.1 |
| Wood Density | AGC ~ Stand Age [a] Wood Density | 4713 | 1.3 | 4.4[a] |
| Prior Land Use | AGC ~ Stand Age [a] Prior Land Use | 1615 | 13.3[c] | 10.0[c] |
| *Management* | | | | |
| Planting Density | AGC ~ Stand Age [a] Planting Density | 1298 | 9.0[b] | 8.4[b] |
| Fertilizer | AGC ~ Stand Age [a] Fertilized | 977 | 5.7[b] | 1.0 |
| Irrigation | AGC ~ Stand Age [a] Irrigation | 329 | 0.7 | 2.9 |
| Weeding | AGC ~ Stand Age [a] Weeding | 703 | 2.6 | 2.5 |
| Thinning | AGC ~ Stand Age [a] Thinning | 895 | 21.5[c] | 65.8[c] |
| Biome | AGC ~ Stand Age [a] Biome | 4696 | 12.0[c] | 14.2[c] |

[a]Indicates significance at the 0.05 level,
[b]indicates significance at the 0.01 level, and
[c]indicates significance at the 0.001 level.
Aboveground carbon (AGC) and stand age are square root transformed in all models. Significance was determined using two-sided F-tests with Satterthwaite approximations of degrees of freedom.

Oceania and Africa (~5%). The dataset included 240 species from 96 distinct genera of tree; however, 33 genera were poorly represented ($n < 3$). Across all observations, mean stand age was 21.3 years (median of 18 years, range of 1–98 years) and mean aboveground carbon stock was 47.0 MgCha$^{-1}$ (median of 36.4 Mg Cha$^{-1}$, range of 1.0–284.6 MgCha$^{-1}$).

The typology and spatial distribution of global plantation forests are poorly characterized due to (a) proprietary holdings of plantation forest data, and (b) difficulties in differentiating planted from natural forests. However, the 2020 Global Forest Resources Assessment (GFRA) of the United Nations Food and Agriculture Organization (FAO)[16] suggest that China (84.7 Mha; 29%), the USA (27.5 Mha; 9%), Russia (18.8 Mha; 6%), Canada (18.2 Mha; 6%), Sweden (13.9 Mha; 5%) and India (13.2 Mha; 5%) hold the largest extents of global planted forests. Data from these countries were represented well within our database, with observations primarily within China (32%), the USA (12%), Russia (11%), Canada (2%), Sweden (1%), and India (3%). Data from Brazil (4%), the United Kingdom (5%), and Japan (5%) also accounted for significant portions of our database.

Our database captures the genera of tree crop species that dominate plantations globally. Comprehensive data on plantation species by location and percent of global plantation forests is difficult to obtain, but 2006 FAO data suggest that *Pinus*, *Cunninghamia*, *Eucalyptus*, *Populus*, *Acacia*, *Larix*, *Picea*, *Tectona*, *Castanea* and *Quercus* species account for approximately 70% of global planted area[17]. These same ten genera accounted for 81% of our observations and we parameterized nonlinear growth functions for seven of these genera with greater than 100 observations in our database.

**Drivers of aboveground carbon accumulation rates.** To explain variation in biomass accumulation, we tested major biological (genus, endemism, and plant traits), human (prior land use and management practices), and environmental (biome) drivers that may control growth (Supplementary Table 1). Almost all drivers explained significant variation in aboveground carbon accumulation rates (Table 1). The effects of individual genera on growth were highly variable, with an almost three-fold difference between

the slowest growing (*Robinia*, $t = -3.43$, $P = < 0.001$) and fastest growing (*Eucalyptus*, $t = 3.09$, $P = < 0.001$) genus. Note that although *Robinia* is often considered a fast-growing species[18], our *Robinia* observations derived from dryland forest areas of China. Aboveground carbon accumulation rates were 127% higher in plantations with exotic rather than endemic species ($t = 7.26$, $P = < 0.001$). Of the five plant traits tested (leaf type, leaf compoundness, leaf phenology, nitrogen fixation capacity, and wood density), leaf type, nitrogen fixation capacity, and wood density all significantly explained variation in carbon accumulation rates. Prior land use significantly affected the rate of aboveground carbon accumulation, with rates on formerly harvested land ($t = -3.95$, $P = < 0.001$) and pasture ($t = -4.16$, $P = < 0.001$) roughly 75 and 66% of rates on former croplands, respectively. Fertilizer use resulted in a relatively minor increase (4%) in the rate of aboveground carbon accumulation ($t = 1.0$, $P = 0.3$) relative to unfertilized stands, whereas thinning decreased the rate of carbon accumulation by 24% ($t = -8.11$, $P = < 0.001$). Finally, the effect of individual biomes on carbon accumulation was variable but ranged less than two-fold.

Next, to examine the relative importance of the drivers, we ran three "full models" on subsets of the data. Data on management practices were highly limiting, so we ran models with and without subsets of management practices. These models identified stand age, genus, prior land use, and ordinated plant trait data as statistically significant effects across all three models (Table 2). Ordinating the plant trait data produced two axes that explained approximately 53% of the trait data variance. The first axis primarily accounted for the leaf type, leaf phenology, and nitrogen fixation data, whereas the second axis primarily represented the wood density data (Table 2 and Supplementary Fig. 4). For the model that excluded management practices (Full Model 1, Table 2), we identified large positive effects for planting of exotic tree species as well as for most of the biomes (relative to boreal conifer forests). Conversely, we found strong negative effects on the rate of carbon accumulation for the coniferous genera (e.g., *Picea* and *Cryptomeria* relative to *Acacia*) as well as prior land uses/disturbances of fire, harvest, and pasture (relative to cropland). Unexpectedly, we found a non-significant effect for

**Table 2 Results of comparative models across drivers.**

| Driver | Full Model 1 (n = 1406) | | Full Model 2 (n = 1,096) | | Full Model 3 (n = 640) | |
|---|---|---|---|---|---|---|
| | Num. DF | F-Value | Num. DF | F-Value | Num. DF | F-Value |
| Stand age | 1 | 1760.9[c] | 1 | 1493.0[c] | 1 | 650.3[c] |
| Genus | 9 | 7.8[c] | 9 | 8.3[c] | 9 | 6.7[c] |
| Endemism | 2 | 3.1[a] | 2 | 3.3[a] | 1 | 0.2 |
| Prior land use | 3 | 8.8[c] | 3 | 24.3[c] | 3 | 18.1[c] |
| Biome | 10 | 2.2[a] | 9 | 1.2 | 9 | 3.1[b] |
| Leaf type, phenology, & N Fixation | 1 | 21.7[c] | 1 | 20.3[c] | 1 | 7.3[b] |
| Wood density | 1 | 0.8 | 1 | 1.2 | 1 | 5.7[a] |
| Planting density | – | – | 1 | 0.2 | – | – |
| Use of fertilizer | – | – | – | – | 1 | 10.8[b] |

[a]Indicates significance at the 0.05 level,
[b]indicates significance at the 0.01 level, and
[c]indicates significance at the 0.001 level.
Full Models 1–3 testing the relative effects across different types of potential drivers. Significance was determined using two-sided F-tests with Satterthwaite approximations of degrees of freedom. For each model, square root transformed aboveground carbon is modeled as a linear combination of the listed drivers with site ID included as a random intercept. Stand age was square root transformed to linearize its relationship with aboveground carbon.

planting density in Full Model 2 (Table 2). We hypothesize that the effect of planting density may be masked due to size-density trade-offs, particularly for monoculture plantations in which stocking is likely to be optimized for maximal production (see Supplementary Discussion). Finally, for the model including the use of fertilizer (Full Model 3, Table 2), we found a significant positive effect of fertilizer use on growth ($t = 3.3$, $P = 0.001$). The individual effect of planting exotic species was not significant in this model ($t = 0.4$, $P = 0.4$), whereas planting exotics had a strong positive effect on rate of carbon accumulation in the other two models.

**Nonlinear growth functions.** We modeled aboveground carbon accumulation using the Chapman-Richards growth function, which is widely used and provides biologically meaningful growth parameters[19–21]. Prior to fitting the function, we aggregated data by plant functional type (e.g., tropical broadleaf species) and genus. Relationships between aboveground carbon and stand age varied across plant functional types and genera (Figs. 2–5). Of the four plant functional types considered, tropical broadleaf forests had the fastest growth rate ($k = 0.24$), which was roughly 2.5 times that of the next fastest growing plant functional type (temperate broadleaf forests) (Fig. 2, 3). Compared to growth rates, asymptotic growth limits varied less and ranged from 73.9 (boreal needleleaf) to 121.0 (tropical needleleaf) MgCha$^{-1}$. Growth rates across the nine genera of tree differed even more than across plant functional types, with the fastest rate (Acacia, $k = 0.33$) roughly 9-fold greater than the slowest rate (Picea, $k = 0.04$) (Fig. 2, 3). Broadleaved genera commonly grown on short rotations (Acacia, Eucalyptus, and Populus) had rapid growth rates, roughly three times those of the coniferous genera (Fig. 2, 4). Lastly, Picea had the highest asymptotic aboveground carbon, which was three times that of Acacia, the genera with the lowest asymptotic aboveground carbon.

The results of the model validation procedure suggest large scatter in the data that is not fully captured by growth functions based solely on age. Normalized RMSE values ranged from 0.74 (Acacia) to 2.58 (Picea), suggesting that model uncertainty ranged from roughly 74% in the best case to 258% of mean aboveground carbon in the worst case. Despite the high uncertainty, the Chapman-Richards growth function is biologically appropriate and is an improvement over commonly used linear growth rates, which do not reflect how forests develop with time, and logarithmic growth functions, which often start with biologically impossible negative intercepts.

**Annualized aboveground carbon accumulation rates.** Although the Chapman-Richards function best captures stand development through time, practitioner and policy audiences frequently employ annual linear rates to inform reforestation planning. We therefore derived annualized aboveground carbon accumulation rates, which varied by genus of tree, plant functional type, and biome (Fig. 6). Broadleaved tropical genera (Eucalyptus and Acacia) had the highest accumulation rates (7.78 ± 0.20 MgCha$^{-1}$yr$^{-1}$ for Eucalyptus) and drove the high mean growth rates for the tropical broadleaf plant functional type and the tropical biomes (Fig. 6a). All four coniferous genera had roughly similar carbon accumulation rates, with the highest mean rate found for Cryptomeria (2.76 ± 0.17 MgCha$^{-1}$yr$^{-1}$). Across plant functional types, differences in aboveground carbon accumulation rates were less pronounced with the exception of tropical broadleaf plantations having the highest rate (6.25 ± 0.17 MgCha$^{-1}$yr$^{-1}$; Fig. 6b). Plantations in tropical & subtropical grasslands, savannas, & shrublands had growth rates (8.18 ± 0.43 MgCha$^{-1}$ yr$^{-1}$) roughly four times those of the slowest growing biome (temperate coniferous forests; 1.62 ± 0.14MgCha$^{-1}$yr$^{-1}$) (Fig. 6c).

Comparing our accumulation rates against those of naturally regenerating forests helps situate our findings but should be done with the understanding that planted and naturally regenerating forests are functionally different systems and carbon accumulation rates are only one metric of comparison[10]. Requena-Suarez and colleagues[22] estimate carbon accumulation rates in younger (<20 years) naturally regenerating secondary forests of approximately 1.1–3.6 MgCha$^{-1}$yr$^{-1}$. Similarly, Cook-Patton and colleagues found annualized mean aboveground carbon accumulation rates to range from 0.1–6.0 Mg Cha$^{-1}$yr$^{-1}$ for natural regeneration[23]. Using the same age classes and spatial aggregations (i.e., the FAO Ecozones), we determined a range of 0.9–8.2 MgCha$^{-1}$yr$^{-1}$ for aboveground carbon accumulation in young plantations. While the lower rates are similar, the high end of the accumulation rates in plantation forests is roughly 1.4–2.3 times greater than those of naturally regenerating forests.

The IPCC also reports mean annual carbon accumulation rates for plantations in Table 4.10 of the 2019 Refinement to the 2006 IPCC Guidelines for National Greenhouse Gas Inventories[24]. However, data are limited (n = 86) and reported for only three genera: Eucalyptus (n = 22; range: 1.4–11.8 MgCha$^{-1}$yr$^{-1}$), Pinus (n = 17; range: 1.2–9.4 MgCha$^{-1}$yr$^{-1}$), and Tectona (n = 7; range: 0.9–7.1 MgCha$^{-1}$yr$^{-1}$). Our carbon accumulation estimates for these three genera align well with the IPCC data, and our database greatly expands (by a factor of ~60) the scope and

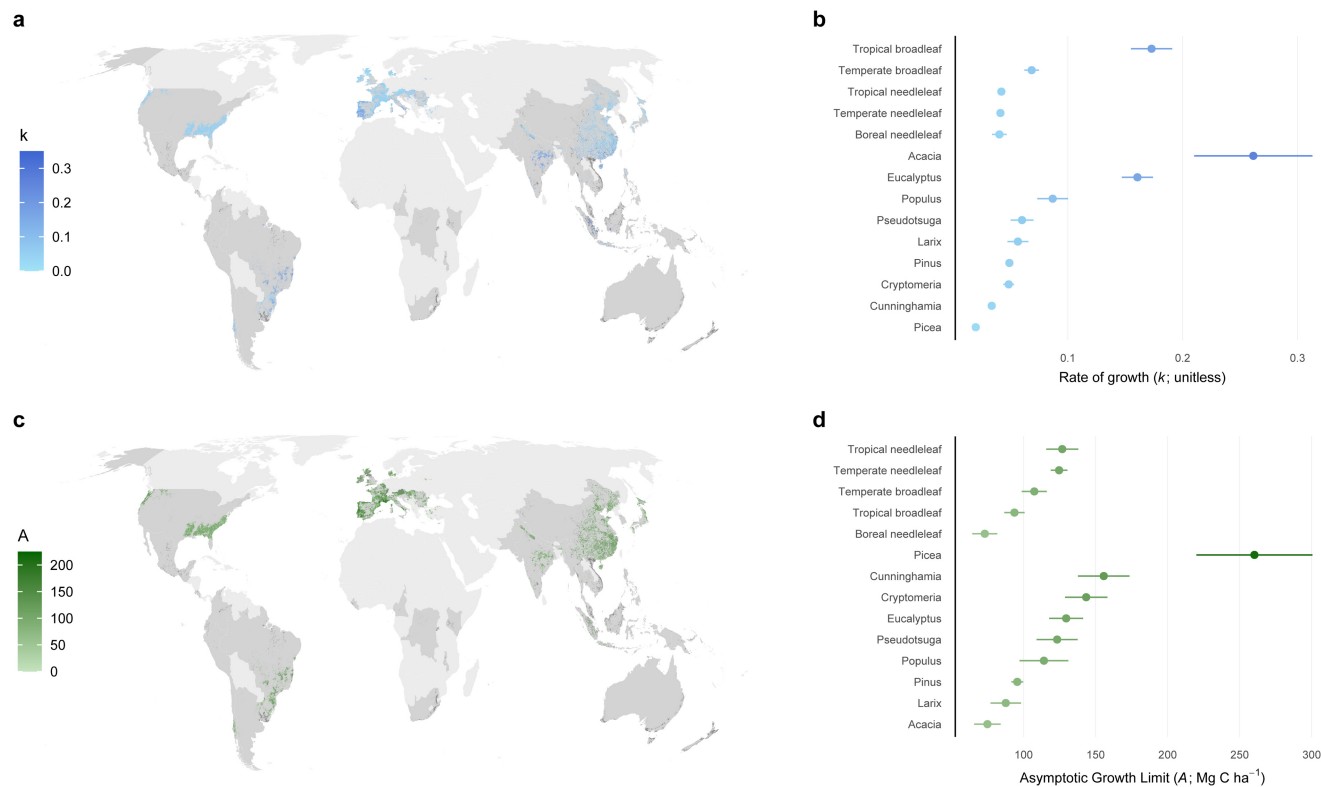

**Fig. 2 Variation in aboveground carbon accumulation across global monoculture plantations.** Variation in growth rates (**a**, **b**) and asymptotic growth limits (**c**, **d**) of global timber and wood fiber plantations. The locations of timber and wood fiber plantations are taken from the Spatial Database of Planted Trees[63]. Genus level growth rates and limits are shown preferentially over those for plant functional type. Plantations with unknown species or plant functional type composition are colored dark grey. Countries in light grey did not have information on spatial location of plantations due to a lack of data collection or the absence of plantations. The points in **b** and **d** are the estimated model parameter values, and the standard errors of the model parameters are shown as error bars. The number of observations associated with the model parameters and standard errors vary by plant functional type and genus and are given in Figs. 2 and 3.

breadth of carbon accumulation observations in global mono-culture plantations.

**Key predictors of above carbon accumulation.** Genus, prior land use, and plant traits explained variation in aboveground carbon accumulation rates. The significance of genus and plant traits suggests that biological drivers primarily explain variation in plantation carbon accumulation at global scales. However, species choice can also be considered a key management decision (i.e., a human factor). Endemism of tree species, another decision-point in species selection, was found to be highly significant until we accounted for fertilizer use. This suggests that if nutrient limitations within plantations are lifted, endemic species may perform as well as exotic species; however, additional work is needed on this topic.

Prior land use, which may be a proxy for site productivity, was also a significant predictor of carbon accumulation. Plantations established on former croplands had higher carbon accumulation rates relative to former pastures, clear-cuts (of both prior rotations and native vegetation), or areas that had burned. This finding parallels the results of others, which show secondary forest growth to be higher on former croplands than pasture, and negative growth effects associated with frequency of fire[25]. While our data do not elucidate a mechanism for this result, hypothesized factors may include soil fertility and/or competition from native vegetation. Our findings further indicate that monoculture plantations established on former crop or pasture lands may accumulate carbon faster than areas that were formerly forested but clear-cut. This finding has key implications for siting

of future plantations, suggesting that clear-cutting of intact forests will not only adversely impact biodiversity, but may also result in lower growth rates than establishing plantations on former croplands or pastures.

Although our results suggest that management practices are important for explaining variation in growth, data limitations constrained our ability to examine the relative effects of management practices. Our interpretation is that while management practices are important for understanding variation in carbon accumulation in plantations, they are difficult to generalize and may therefore have limited utility in predicting carbon accumulation across broad scales.

**Nonlinear accumulation of aboveground carbon in plantation forests.** Our use of nonlinear growth functions provided biologically meaningful understanding of aboveground carbon accumulation in monoculture plantations. For example, we estimated rapid carbon accumulation rates for genera commonly grown on short rotations for pulpwood (e.g., *Eucalyptus*, *Acacia*, *Populus*) versus slower carbon accumulation rates for coniferous species that are commonly grown for timber (e.g., *Picea*, *Cunninghamia*, *Pseudotsuga*). Although this is an expected result, our parameterized growth functions are valuable for multiple reasons (Supplementary Tables 2, 3). First, they are significant improvements over annualized mean aboveground carbon accumulation rates, which do not account for how carbon accumulation varies with time. This distinction is critical for accurately assessing time-dependent carbon accumulation within the forest sector. For example, the delayed carbon accumulation rates that occur during

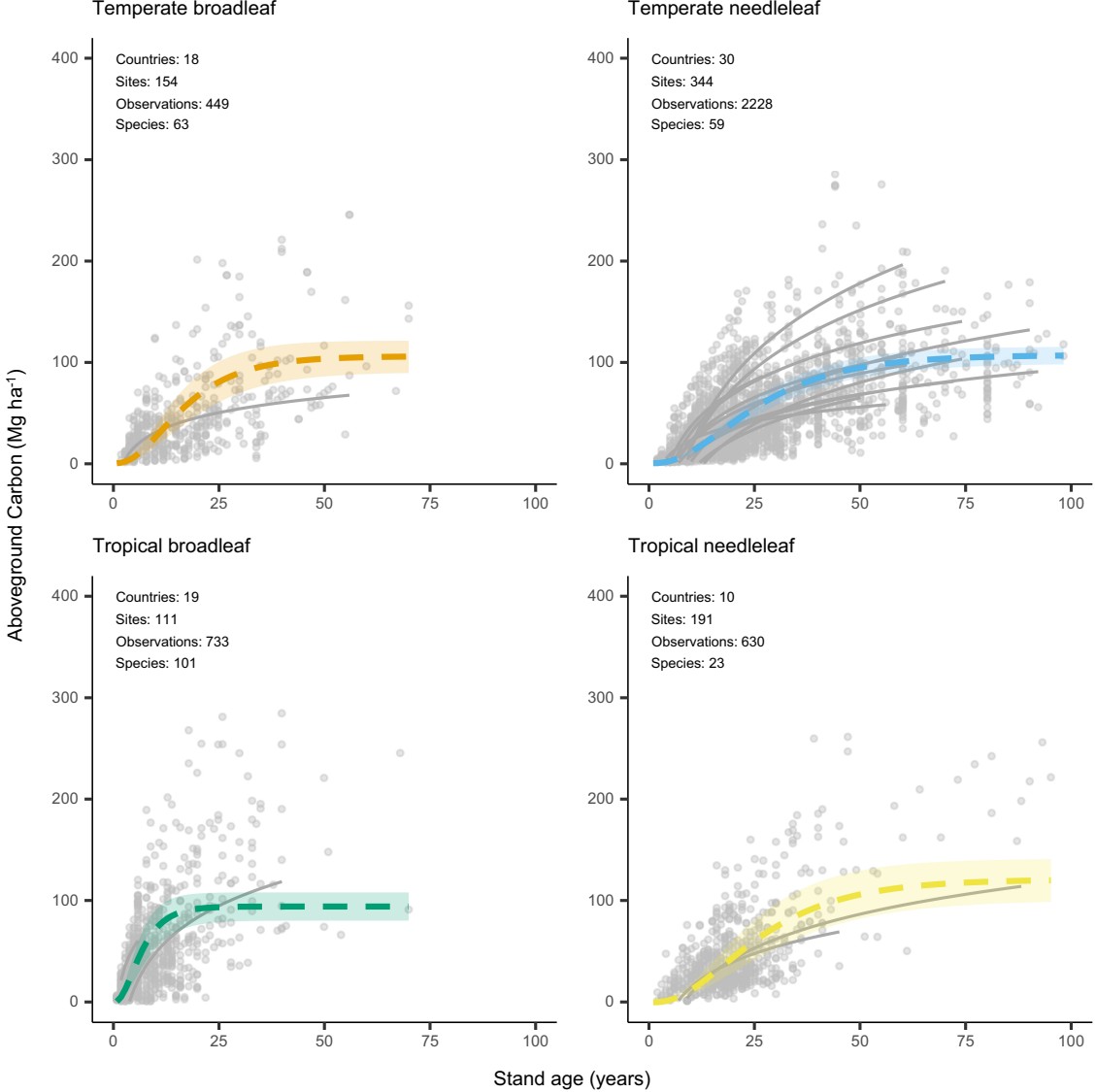

**Fig. 3 Growth functions by plant functional type.** Parameterized Chapman-Richards growth functions, with the shaded areas around curves denoting 95% confidence interval of predictions based on fixed effects only. The grey curves (for visual comparison of species level trends) are logarithmic relationships between stand age and aboveground carbon for individual species with greater than 40 observations in the database. The growth function for the boreal needleleaf plant functional type is shown in Supplementary Fig. 5.

stand initiation (e.g., ~5 years) represent half of the time window under which many public and private-sector programs have committed to reduce emissions (e.g., by 2030). Assumption of immediate and sustained rates of carbon accumulation within forestry projects may systematically overestimate climate benefits at decadal scales.

Second, plantation forestry is an attractive mitigation strategy for climate change given the economic benefits that can accrue to landowners, which may help incentivize restoration of forest cover[26]. Accounting for temporal variation in growth is important for accurately modeling economic returns from plantation forests. For example, forest management decision-making such as identifying optimal rotation lengths (including for joint management of timber and carbon) employs nonlinear modeling of stand growth[27]. Although a range of growth models exist (e.g., see the FORMODELs database), they tend to be developed for site to regional level scales. Our dataset and growth functions complement these models by providing an open-source and freely available framework for modeling carbon

accumulation in the dominant monoculture plantation types seen across the globe.

**Caveats and potential limitations.** Several important caveats and limitations to our analyses exist, which we discuss here to facilitate appropriate use of our results. First, we lack clear understanding of the global distribution and type of monoculture plantations, making it challenging to assess whether our dataset represents the full range of variation in carbon accumulation seen across the globe. However, comparing our dataset to the most comprehensive overview of monocultures (i.e., the 2020 Global Forest Resources Assessment) suggests that most of our data are from those countries and regions that dominate global plantation forestry. Furthermore, grouping the data by biome, genus, and age class and examining the number of observations per site shows balance within the dataset (mean number of observations per site = 2, median = 1, max = 15, or less than 0.5% of all data; see Supplementary Information). To the best of our knowledge,

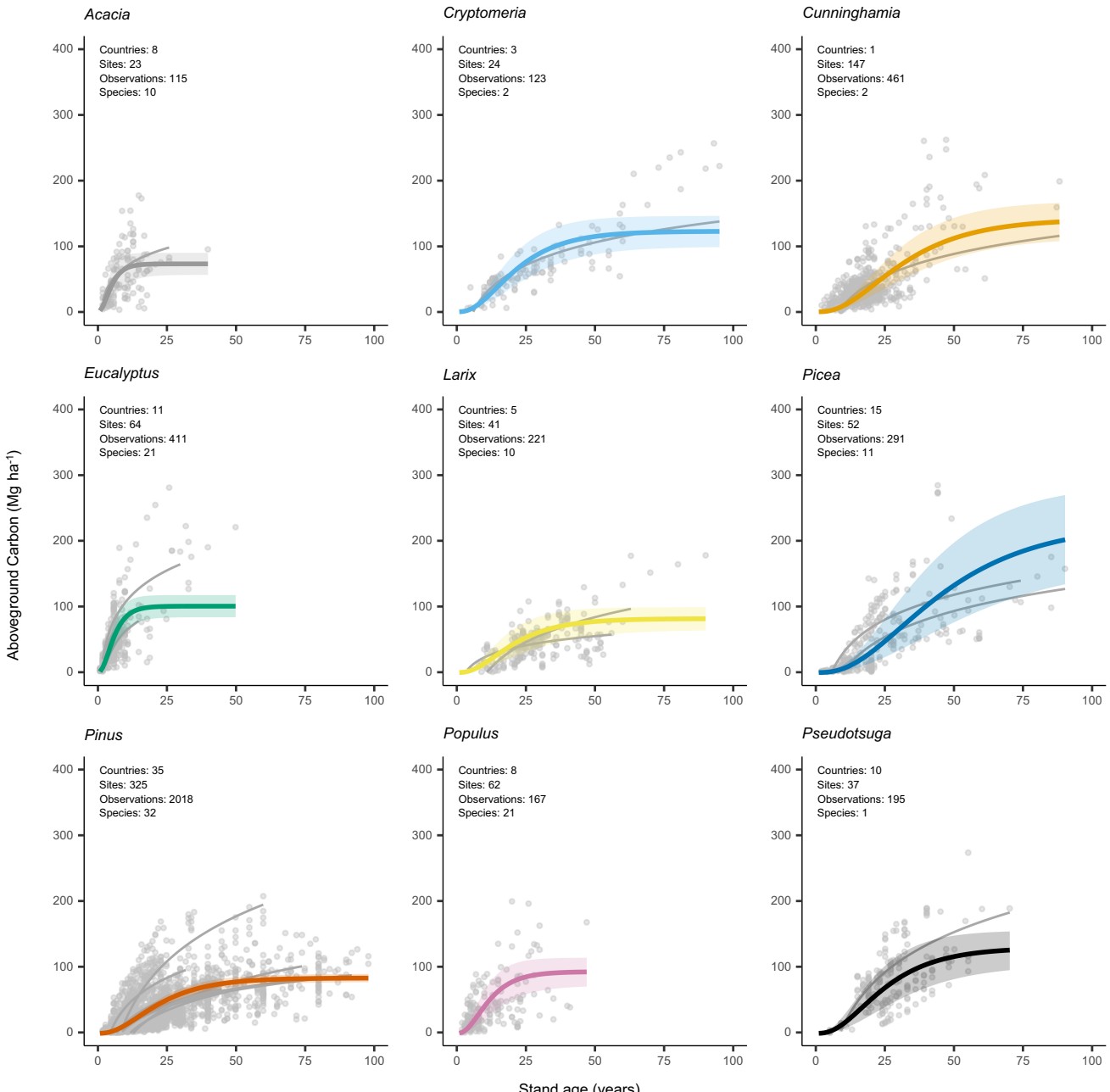

**Fig. 4 Growth functions by genus.** Parameterized Chapman-Richards growth functions, with the shaded areas around curves denoting 95% confidence interval of predictions based on fixed effects only. The grey curves (for visual comparison of species level trends) are logarithmic relationships between stand age and aboveground carbon for individual species with greater than 40 observations in the database.

our dataset is the most comprehensive compilation of carbon accumulation in monoculture plantations in the public domain.

Our asymptotic growth limit estimates (particularly the $A$ parameter) should be interpreted with caution. The data we compiled are primarily from plantations managed on economically driven rotation ages and do not represent the maximum carbon stocks that could be achieved if these plantations were left to grow in perpetuity. Rather, the asymptotic values we present are fixed effects estimates for the population-level mean aboveground carbon stocks across all sites. Differences in site class are captured within our random effects estimates. For example, when accounting for these site level effects, the maximum predicted aboveground carbon value of our *Pinus* model was 164 MgAGCha$^{-1}$, roughly double that of our population level *Pinus*

asymptote (see Supplementary Information). However, these random effects cannot be estimated for sites in which we have no training data, and fixed effects estimates of the population-level parameters are most appropriate for approximating aboveground carbon accumulation at new sites. Further, the potential risks of biased asymptotic growth limits can be mitigated by restricting use of our parameters to near decadal scales (e.g., ~ 20–30 years). Thus, our growth parameters are perhaps most relevant for the next few climate-critical decades.

The Chapman-Richards growth function is one of many within the literature and our parameter estimates may be sensitive to the selection of this function. A key criticism of the Chapman-Richards function is instability in the model parameters[19]. However, others have examined the risk of parameter instability

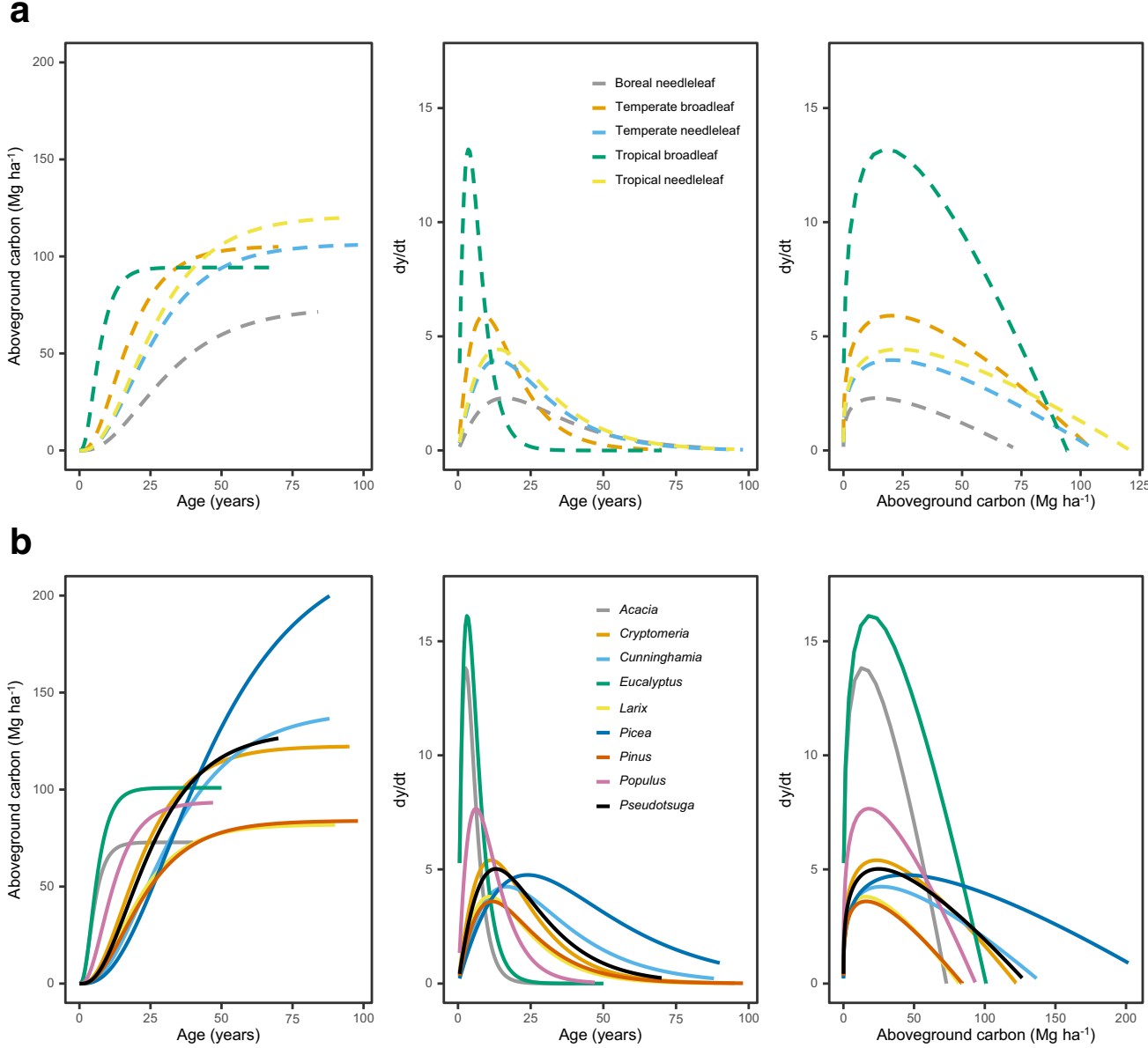

**Fig. 5 Comparison of aboveground carbon accumulation and growth rates across plantation types.** Aboveground carbon accumulation as a function of age, growth rate as a function of age, and growth rate as a function of stand level aboveground carbon are shown for different plant functional types (**a**) and genera (**b**). The growth rate (dy/dt) is calculated using the differential form of the Chapman-Richards function. All model parameters were estimated with the *m* parameter of the Chapman-Richards growth function fixed at 0.67 (i.e., von Bertalanffy special case of the Chapman-Richards growth function).

when applying Chapman-Richards growth functions to aggregated datasets and have shown that, for the relatively constrained system of even-aged monoculture plantations, the Chapman-Richards growth function performs well[28]. Rather, parameter instability in the asymptotic growth limit estimate is a greater issue for multi-species and multi-aged natural forests, as well as using the parameterized functions to model growth beyond the age range of the data. Further, others have concluded that the biological basis of growth functions have been overstated and users should focus instead on plausible growth functions that best meet their needs, as well as parameter estimation strategies[29]. Here, we elected to use a theoretical growth function that is among the most widely used within forestry and empirically derived parameter estimates using a global dataset. By making our dataset publicly available, we anticipate that others will investigate

the effects of alternative functional forms on the parameter estimates.

Lastly, we did not incorporate changes in belowground biomass and soil organic carbon into our analyses, but discussion of carbon flows to and from these pools is warranted. For belowground biomass, we excluded this pool given little data and constraints associated with synthesizing across studies with highly variable sampling protocols[30]. Instead, we recommend using root:shoot ratios, which are available for particular forest types and regions[31], as well as through spatially explicit maps at global scales[32]. These ratios allow carbon stocks in belowground biomass to be modeled as a function of the aboveground carbon values obtained from our models. Soil organic carbon, often the largest carbon pool in forested ecosystems, is impacted by forest management practices in varied ways[33]. Generally speaking, only

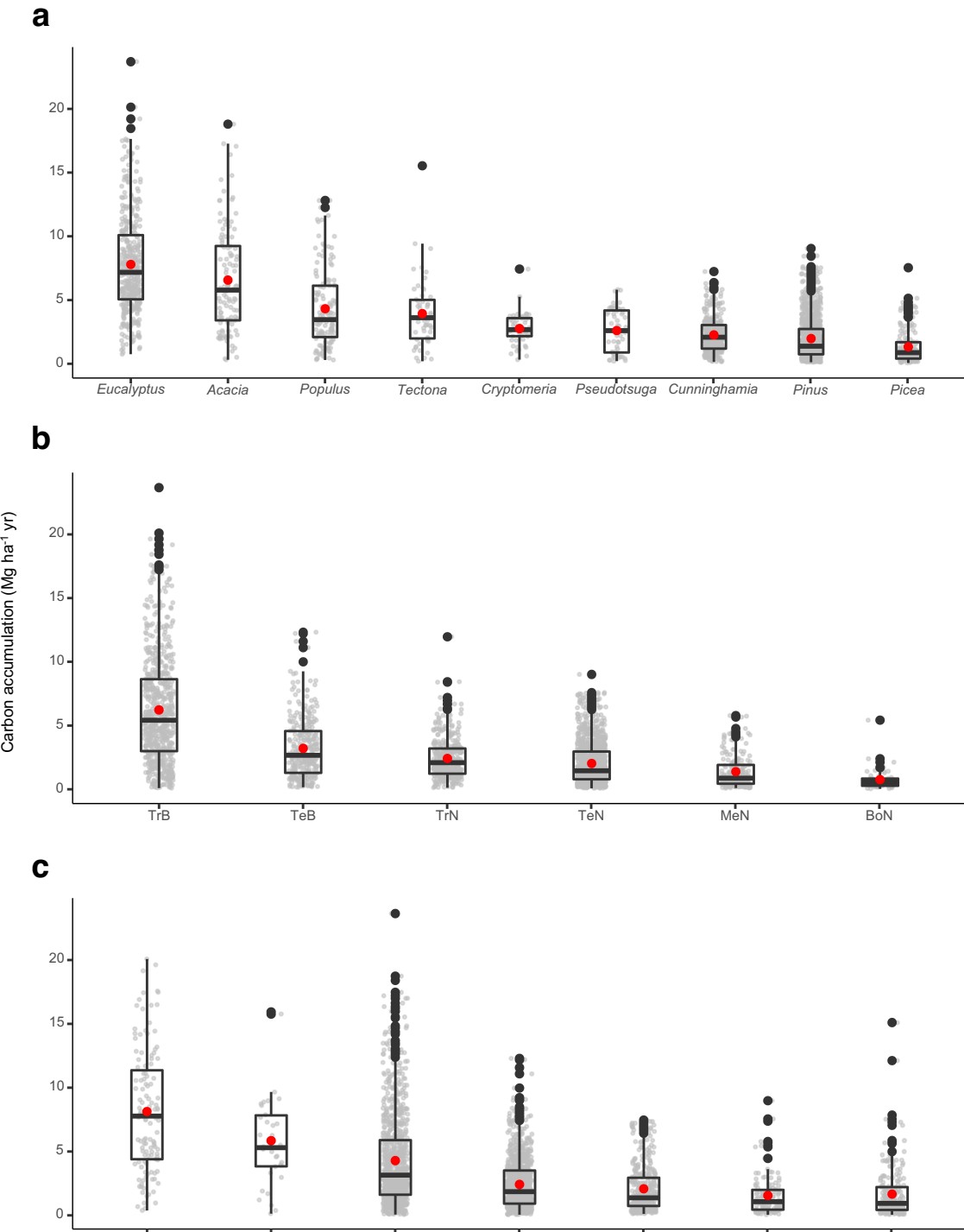

**Fig. 6 Annualized aboveground carbon accumulation.** Annualized aboveground carbon accumulation rates for the major (**a**) genera, (**b**) plant functional types, and (**c**) biomes. Plant functional type codes: TrB – tropical broadleaf; TeB – temperate broadleaf; TrN – tropical needleleaf; TeN – temperate needleleaf; MeN – Mediterranean needleleaf; BoN – boreal needleleaf. Biome codes: TSGSS – tropical & subtropical grasslands, savannas, and shrublands; TSDBF – tropical & subtropical dry broadleaf forests; TSMBF – tropical & subtropical moist broadleaf forests; TBMF – temperate broadleaf & mixed forests; MFWS – Mediterranean forests, woodlands & scrub; TCF – temperate conifer forests; TGSS – temperate grasslands, savannas & shrublands. Boxplots indicate the median values and the 25th and 75th percentiles of the data, whereas the whiskers extend to values no further than 1.5 * the interquartile range. The red points represent mean values. The number of observations for each grouping are: Eucalyptus – 397; Acacia – 110; Populus – 132; Tectona – 54; Cryptomeria – 52; Pseudotsuga – 56; Cunninghamia – 321; Pinus – 1,072; Picea – 177; TrB – 656; TeB – 298; TrN – 374; TeN – 1,076; MeN – 213; BoN – 54; TSGSS – 123; TSDBF – 34; TSMBF – 879; TBMF – 929; TGSS – 346; TCF – 134; MFWS – 232.

the forest floor and the upper soil horizons (e.g., O and A horizons) are expected to be impacted, with recent reviews estimating that <10% of total soil organic carbon stocks might be impacted[33]. The magnitude of impact will depend upon the type and intensity of management practice (in particular, harvesting and site preparation), with more intense soil disturbance inducing greater soil losses.

Future assessments of the climate change mitigation potential of monoculture plantations would do well to incorporate these belowground carbon pools. Moreover, additional empirical data is needed to improve the representativeness and balance of the dataset we have compiled here. Specifically, we recommend that future studies focus on i) notably data-poor regions (e.g., Africa), ii) repeatedly measured plots, iii) better data sharing, and iv) standardized metrics (e.g., age, biomass, site index) to facilitate synthetic understand of carbon accumulation in plantations[34]. In doing so, we can improve our understanding of how carbon accumulates in plantation forests across the globe and facilitate assessments of the full climate impacts of plantation forests (including both short-lived and long-lived harvested wood products). Lastly, although we did not directly account for mortality here, it will be important to consider how shifts in climate regimes and disturbance patterns present new risks to these systems, particularly for slow-growing plantation species that are investments over approximately century-scale time frames.

**Potential applications of our findings**. Monoculture plantations are only one pathway of many (e.g., natural regeneration, assisted natural regeneration, agroforestry, diverse plantations) for restoring forest cover[2,35]. However, they represent the dominant approach underlying restoration commitments and thus merit careful consideration[4]. When managed sustainably and integrated into a broader landscape, plantation forestry can help reduce pressures on mixed-use forests (e.g., concessions) and meet demand for harvested wood products efficiently[5]. Substitution of sustainably produced timber for carbon intensive products such as concrete or steel can also reduce embodied carbon and expand carbon stocks beyond what exists on the landscape[36,37]. However, emissions from harvesting activities, decomposing harvest residues, and short-lived wood products may reduce or negate these benefits[38]. To accurately assess the climate benefits of plantations, enthusiasm for the high early growth rates we quantified here must be tempered by careful life cycle accounting. Furthermore, accounting for potential changes in the belowground biomass and soil carbon stocks will give a more holistic picture of the climate change mitigation potential of plantations.

Our results also have important implications for species choice. Exotic monocultures generally have poor biodiversity outcomes compared to other forest types[39]. Concerning declines in biodiversity[40] coupled with global momentum to conserve biodiversity (e.g., the UN Decade on Ecosystem Restoration) encourage interventions to the biodiversity and climate crises in tandem. Using endemic species in monocultures instead of exotics could improve biodiversity outcomes, and the higher carbon accumulation rates that we observed in exotic monocultures disappeared after accounting for fertilizer use. This suggests that careful nutrient management in endemic plantations could help to optimize both climate and biodiversity benefits. However, further research on the trade-off between fertilizer use versus exotic species would be of high value, since fertilizer can negatively impact the local environment[5] and result in additional climate emissions[41].

We also identified traits linked to higher carbon accumulation rates, which can be used to select additional species not commonly used within plantation forestry. A handful of species and genera dominated our dataset and these were generally not endemic to the places where they were grown. Identifying additional endemic species to use in lieu of exotic species would improve the site-level biodiversity value of plantation forests[42], and investing in plantation polycultures would help to resist trends towards global biotic homogenization[43]. Biodiverse forests are also likely to be more resilient to disturbances such as pests or natural disasters, which is key for the durability of carbon sequestered in natural ecosystems, particularly under a changing climate[44]. Moreover, species-rich forests may store more carbon than species-poor forests due to complementary resource use (niche complementarity)[45,46]; however, competitive and mutualistic interactions among species may vary[47,48]. Additional work is urgently needed to better understand how diversifying plantations could improve climate and biodiversity outcomes.

## Methods

**Dataset compilation & standardization**. We systematically reviewed the literature to identify studies reporting data on biomass and carbon stocks in monoculture plantations. The search was part of a larger effort to quantify biomass accumulation associated with re-establishing tree cover more generally[23]. We describe the dataset compilation and standardization process here, with additional details in the Supplementary Methods and Cook-Patton et al.[23].

The literature search considered over 11,000 articles, which we filtered to 640 studies quantifying biomass (or carbon) stocks in monoculture forest plantations. We then further winnowed these to include studies that reported (i) empirical measurements of biomass or carbon in the aboveground pool; (ii) age of the plantation at the time of field measurements; and (iii) a latitude and longitude pair or sufficient geographic detail from which geographic coordinates could be obtained. We only collected data on aboveground stocks given that belowground biomass and soil organic carbon data were sparse, and variation in field sampling methods make synthesizing belowground biomass and soil organic carbon stocks across studies difficult[23,30,49]. We omitted understory biomass given that understory vegetation in monocultures is a small proportion of total biomass[10]. Lastly, we only considered live biomass. We did not consider dead wood which is likely removed from many systems via precommercial thinning and was a relatively small proportion of total aboveground carbon stocks.

For each included study ($n = 424$), we collected information on biomass (or carbon) stocks, age, geolocation, tree crop species, prior land use/disturbance, and management practices such as planting density, rotation length, site preparation, fertilization, irrigation, vegetation control, and thinning. To account for spatial structures in the data, we grouped measurements by site. When studies empirically determined the percent biomass that was carbon, we retained their values; however, when studies used a default factor (e.g., 0.5) to convert biomass to carbon, we adjusted these values using the Intergovernmental Panel on Climate Change's recommended factor of 0.47[24].

**Potential drivers of variation in aboveground carbon accumulation**. To explain variation in aboveground carbon accumulation across plantation types, we sought to account for the major biological (genus, endemism, and plant traits), human (prior land use and management practices), and environmental (biome) drivers that may control growth. We describe the collection of these data below and summarize key characteristics of each of the potential drivers in Supplementary Table 1.

Tree crops are commonly selected from a limited number of genera for characteristics such as growth rate and suitability for wood products. We collected data on the planted tree species from all studies; however, we collapsed the species data to genus (including both hybrids and clones) to reduce the feature space of the dataset. Additionally, to test for the effect of planting exotic species on growth, we coded all species as being either endemic or exotic to the locale in which they were planted.

Although genus can serve as a proxy for suites of plant traits, we further examined the effect of species-level plant traits given that trait data may vary within genera. To create an initial list of candidate traits, we used the TRY Plant Trait Database to assemble trait data for as many of the planted species in our database as possible[50]. Although we identified 12 candidate traits, we ultimately excluded eight traits that did not have sufficient variation across species. For example, all species were classified as "woody" for the "plant woodiness" trait. For the traits we included (leaf compoundness, leaf phenology, leaf type, and nitrogen fixation capacity), we then used species descriptions compiled by Botanic Gardens Conservation International (e.g., the Global Biodiversity Information Facility) to manually fill data gaps for species not found in TRY. Additionally, we extracted species-specific wood density data from the Global Wood Density Database[51]. We averaged wood density values for species with more than one wood density

observation in the database, whereas we used genus level averages for species not found in the database.

Prior land use can control subsequent vegetation growth depending on both the type and intensity of land use[52]. Studies commonly reported prior land use type, which we coded into our database. However, we were unable to include intensity of land use as it was rarely reported. We generally found four types of land use/ disturbance to be well-represented within the literature: clear-cut harvest, croplands (inclusive of shifting cultivation), pasture, and fire. When studies reported multiple prior land use types, we recorded the most recent type.

Plantations often receive active management to optimize growth. We examined the effect of management practices on carbon accumulation to the degree that studies allowed. Studies did not consistently report management practices at the same level of detail, but many provided information on planting density, site preparation, fertilization, irrigation, management of competing vegetation, or thinning. We qualitatively recorded all management practices that were reported in the studies, which we then coded into a presence vs. absence variable for statistical analyses. Although substantial variation in the use of a given management practice may exist (e.g., different types or quantities of fertilizer), a presence/absence variable for each type of management practice was an optimal compromise between accounting for the use of management practices and obtaining management data for as many studies as possible.

Finally, biomass accumulation in plantations is expected to vary across climates, which can be proxied by biome type. To examine the effect of climate on plantation biomass, we spatially intersected the locations of all sites with maps of major ecological zones. Namely, we used both a map of global biomes as well as the United Nations Food and Agriculture Organizations classification of ecozones[53–55]. We refer to the first as "biomes" whereas we refer to the second dataset as "FAO Ecozones." Plantations are not common across all global biomes and we therefore omitted biomes for which data were sparse (e.g., flooded grasslands and savannas, mangroves, and montane grasslands and shrublands). Further, plantations in non-forest biomes are often not successful and can cause severely negative biodiversity impacts[6].

**Statistical analyses of variation in aboveground carbon accumulation.** For each of the potential drivers of aboveground carbon accumulation in plantations, we tested for their effect on plantation carbon using linear mixed effects models[56]. We square root transformed both aboveground carbon and stand age prior to fitting the models to improve linear relationships (Supplementary Fig. 1). We then modeled aboveground carbon as a function of stand age, the driver under consideration, and the interaction of stand age and the driver (all fixed effects). We included random intercepts for site in all models to account for spatial autocorrelation[57]. Since data on drivers were missing for some observations, we subsetted the data to each driver of interest before fitting the models. When testing genus of tree crop, we filtered the subsetted dataset to only genera with 20 or more observations to reduce the levels of the driver and potential effects of sparse data. Lastly, when testing the categorical management drivers (e.g., use of fertilizer), we filtered the subsetted dataset to only genera with observations across all levels of the driver (e.g., both fertilized and unfertilized) to improve balance across the data.

We tested for the statistical significance of the potential drivers and their interaction with stand age (i.e., a separate model for each driver; see Table 1) using F-tests with Satterthwaite approximations of degrees of freedom and restricted maximum likelihood, as implemented in the {lmerTest} package of Program R[58,59]. Similarly, we examined the significance of individual levels of the drivers using t-tests (and again, Satterthwaite approximations of the degrees of freedom) to obtain p-values. These approaches reduce the likelihood of a Type I error, particularly for models with large numbers of observations such as ours[58].

Next, to examine the relative effects of the drivers, we ran three "full models" using only those observations that had complete data across three subsets of drivers. Data on management practices were highly limiting and we therefore considered only two management practices—planting density and fertilizer use— each of which was included in a separate model. Our first full model (Full Model 1) included all drivers except those related to management (i.e., genus, endemism, plant traits, prior land use, and biome), whereas our second and third models included all these drivers plus each of planting density (Full Model 2) and fertilizer use (Full Model 3). We had to consider these management practices in separate models given that observations primarily reported only one of the practices, but not both. Additionally, the trait data were categorical in nature, repeated across species, and likely exhibited correlation across traits. We therefore ordinated the trait data prior to their inclusion using Factor Analysis of Mixed Data (FAMD), a principal component method for datasets of both continuous and categorical variables. Similar to our models of individual drivers, we filtered the subsetted data to genera with at least 20 observations and included random intercepts by site to account for spatial autocorrelation. We determined significance of the driver fixed effects using the same approach as our individual driver models (i.e., F-tests using the Satterthwaite approach). Given data limitations, we did not include interactions in the model but specified each driver as an additive effect. All models were fit using the {nlme} package in Program R[60].

**Development and validation of growth functions.** We also parameterized non-linear growth functions of aboveground carbon stocks in plantations for all genera

and plant functional types with more than 100 observations in the database. Plant functional types are a means of representing broad classes of plants that share similar growth forms and life histories across biomes. For our purposes, plant functional types are a convenient way of incorporating data from relatively less-represented genera into growth functions, as well as producing models that are generalizable across plantation species not included in our database. Here, we combined climatic information from our biome dataset with leaf type (e.g., tropical broadleaf species) to produce plant functional types for all species within our database.

We considered four different types of growth function to be fit to the data: logarithmic, linearized logistic, logistic, and the Chapman-Richards growth function (Supplementary Fig. 2). After assessing the fit of each function to the data (Supplementary Methods), we elected to fit the Chapman-Richards function using Eq. (1), which is (i) based in theoretical growth of forest stands over time, (ii) more flexible than other logistic functions, and (iii) widely employed within the forest modeling literature[19–21]. The Chapman-Richards growth function is specified as:

$$y(t) = A * (1 - b * e^{(-k*t)})^{(\frac{1}{1-m})} \qquad (1)$$

Where for our particular case, $y(t)$ is aboveground carbon in Mg ha$^{-1}$ at time $t$, $t$ is the age of the forest stand in years, and $A$, $b$, $k$, and $m$ are parameters to be statistically estimated from the data.

The Chapman-Richards growth function has theoretical foundations and its parameters are biologically meaningful[19,61]. Specifically, $A$ represents the asymptotic limit of the response variable, $k$ is the rate of growth, and $m$ is a shape parameter. When primarily concerned with flexible fits to empirical data (i.e., the trademark of the Chapman-Richards growth function), it is best to estimate $b$ and $m$ from the data. However, within explanatory analyses such as ours, it is common to fix $b$ and $m$, allowing only the asymptotic limit ($A$) and growth rate ($k$) to be statistically fit to the data[19].

Given our objectives of comparing aboveground carbon accumulation across plantation types, we followed this procedure and fixed $b$ at 1 [effectively stating that AGC $(t=0) = 0$] and fixed $m$ at 0.67. Fixing $m$ at 0.67 is a common practice within the literature and produces the von Bertalanffy special case of the Chapman-Richards function, which is the original function that Richards generalized[19,29,61]. We therefore fixed $m$ at 0.67; however, we also considered how alternative values of $m$ affected our parameter estimates (see Supplementary Information). After fixing $m$ at 0.67 and $b$ at 1, we statistically estimated $A$ and $k$ for each plant functional type and genus using non-linear mixed effects models, with site included as a random effect. All growth models were parameterized using the {saemix} package in Program R (version 4.0.4), which employs the Stochastic Approximation Expectation Maximization algorithm to derive maximum likelihood estimates of the parameters[62].

We assessed the fit of the models using root-mean-square error (RMSE), calculated via a cross-validation procedure with a 15 to 85% validation data to training data split. Given that we used mixed effects models, we set aside data for validation by randomly selecting all data from 15% of the sites rather than 15% of observations across all sites. Although doing so ensures a true out-of-sample validation, the unbalanced nature of our database caused the number of observations included in the training versus validation datasets to vary. We therefore bootstrapped this procedure a total of 25 times for each growth model, retaining the RMSE values from each run and averaging across them to obtain our final RMSE estimates. We report both the averaged RMSE values as well as RMSEs that are normalized by mean aboveground carbon, which facilitates comparison across models.

**Calculation of annualized carbon accumulation rates.** Finally, to improve the utility of our analyses for broader policy and practitioner audiences, we also generated annualized rates to use as default values of aboveground carbon accumulation in monoculture plantations. We calculated plot-level carbon accumulation rates by dividing stand-level aboveground carbon by stand age. To reduce the effects of different species tending to be managed on different rotation lengths, we first filtered all data to stands younger than 30 years in age. After calculating the rates at the plot level, we subsequently summarized these values for the major i) genera, ii) plant functional types, and iii) biomes in our database. Unless otherwise stated, all values presented in the text are mean values ± the standard error of the mean. Despite providing these annualized rates, we hope the policy and practitioner audiences adopt our more biologically accurate nonlinear representations of carbon accumulation in plantation forests.

**Reporting summary.** Further information on research design is available in the Nature Research Reporting Summary linked to this article.

## Data availability
The monoculture plantations aboveground carbon stock data compiled in this study are published on Zenodo (https://doi.org/10.5281/zenodo.6555216). A publicly facing version of the Spatial Database of Planted Trees is available through the World Resources Institute (WRI), whereas the version used here may be requested through Global Forest

Watch at WRI. The Global Wood Density Database is available through DataDryad (https://doi.org/10.5061/dryad.234). The plant trait data were accessed through the TRY Plant Trait Database (https://www.try-db.org/TryWeb/Home.php). The spatial information on biomes of the globe were obtained through the RESOLVE Ecoregions 2017 database (https://ecoregions.appspot.com/). The Food and Agriculture Organization of the United Nation's Global Ecological Zones are available through the FAO's data center (https://data.review.fao.org/map/catalog/srv/search?keyword=FRA). J.J.B welcomes discussions around potential collaborations in using and expanding the data published here.

## Code availability

All code for the models and driver analysis of this manuscript are available on GitHub (https://github.com/jbukoski/GPFC) and have been archived on Zenodo (https://doi.org/10.5281/zenodo.6588710).

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

## Acknowledgements

We thank Greg Biging, Bodie Cabiyo, Iryna Dronova, Karen Holl, Jeff Vincent, and the Potts Group at U.C. Berkeley for valuable feedback on early drafts of this manuscript. Although their comments have greatly improved our study, the ideas and opinions expressed in our study do not necessarily reflect their own. We thank the Children's Investment Fund Foundation, COmON Foundation, the Doris Duke Charitable Foundation, and Good Energies Foundation for support of the initial literature search. We thank the Bezos Earth Fund for supporting S.C.C.-P.'s time on this work. J.J.B. was supported by a Berkeley Connect fellowship while conducting this work.

## Author contributions

J.J.B., S.C.C.-P., and M.D.P. designed the study. J.J.B. led the writing of the article, with substantial contributions from S.C.C.-P. and M.D.P. J.J.B., S.C.C.-P., C.M., H.B., and J.L.C. reviewed the literature and compiled the monoculture aboveground carbon observations. E.D.G., and N.L.H. contributed data on the locations of existing plantation forests. J.J.B. performed all data analysis and produced all data visualizations. All authors revised the article.

## Competing interests

M.D.P. is the Chief Science Officer and J.J.B. is a Scientific Advisor for Carbon Direct Inc., a company combining science, technology, and capital to deliver quality $CO_2$ management at scale. M.D.P. is a shareholder in the company and thus stands to benefit financially from forest management targeted at climate change mitigation. The remaining authors declare no competing interests.
