## [Peer Review File · Nature Communications]

Response to reviewer comments, first round –

Reviewer #1 (Remarks to the Author):

The revised manuscript is markedly improved and the expanded descriptions and discussion on limitations of the study are valuable. A few additional considerations. First, is still not clear what the accumulation rates and growth models include. I believe it is aboveground live tree biomass carbon. If so, it would be helpful to spell that out in the manuscript (and clarify in the title) and since mortality was not included in the growth model development that the estimates reflect gross accumulation. Second, while you mentioned repeat measures in your discussion of future assessments and the limitations of the data compiled for use in the analysis, it would be worth expanding on the use of repeat measures as a means to empirically obtaining rates of change in live trees and also tree status, particularly given changes in the drivers of tree growth and mortality. Finally, while the study is global in scope I am still not convinced that there is anything particularly novel about the methods or analysis that warrant consideration in *Nature Communications* and I would suggest the authors consider a more specialized outlet.

Reviewer #2 (Remarks to the Author):

REVIEW OF THE MANUSCRIPT 348060_0_art_file_6181187_r55x7g (previously submitted to *Nature Climate Change* as 23668_0_art_0_qvb1f9)

Rates and drivers of aboveground carbon accumulation in global monoculture plantation forests (in the previous submission it was Rates and drivers of carbon accumulation in global monoculture plantations)

by

Bukoski et al.

The authors present a study that has the objective to estimate aboveground carbon accumulation in global monoculture plantations. Their objective is interesting of course and the article is original and the results are relevant.

I had already reviewed a previous version of this manuscript for a related journal. The authors considered most of my previous comments/suggestions as well as those from reviewer #1, namely: 1) they are more modest in what concerns objectives and added a good discussion about the limitations of the study (pointed out by the reviewers in relation to the previous submission); 2) they include the data (and also the scripts), therefore we can check them in more detail.

However, I think that there are some issues that need to be better explained:

Data in the csv file should have a description of the variables (some I don't even know what they are and most do not have units) and the references of the papers from which the data were extracted should be provided. I am familiar with some of the data and the above ground carbon values look strange, I would like to check from which publication you got the information. And this may be true for many other readers.

I still have a problem with the interpretation of the asymptote estimates as they are biased by the young ages of the plots measured for some species. But the authors discuss this problem in the discussion section, so it is better than in the previous submission.

Lines 122-134 – it is not clear to which statistical test the authors are referring to when they mention the drivers of aboveground carbon accumulation rates, and I was not able to deduct it from the information provided in the Methods section. Maybe the t-tests for the significance of the parameters in the linear models? Please improve the quality of these paragraphs

Line 147 – it is really strange that planting density does not have an effect, I think the problem must be that there are several multicollinearities among the different drivers and you tested one driver at a time added to a model with age. The impact of planting density may be masked by other driver(s). I "played" a bit with the data and fixing the species and the ages, we can see some increasing tendency with planting density. Therefore, fitting linear mixed models of aboveground c accumulation as a function of just age and one driver may not be a good option. Please discuss this detail.

Line 500 – I suppose it is better to say "... functional types with more than 100 observations..." than "... functional types with greater than 100 observations..."

Lines 522-528 – about fixing the b and m parameters of the Richards function: I agree with fixing $b=1$ as it just indicates that $abc=0$ when $t=0$, which is fine, but the m parameter (I would not call it an allometric constant, it is a shape parameter) is related to the shape of the curve, namely with the location of the inflection point, I don't see a reason to fix it. You can go to <http://home.isa.utl.pt/~joaopalma/modelos/fgfp/> and check the important role of m in the shape of the curve (and, to a certain, extent, in the growth rate).

REVIEWER COMMENTS

Reviewer #1 (Remarks to the Author):

The revised manuscript is markedly improved and the expanded descriptions and discussion on limitations of the study are valuable. A few additional considerations.

Thank you for noting that our manuscript is improved and the discussion on limitations is valuable. We feel that the feedback from the reviewers has substantially improved our manuscript.

First, is still not clear what the accumulation rates and growth models include. I believe it is aboveground live tree biomass carbon. If so, it would be helpful to spell that out in the manuscript (and clarify in the title) and since mortality was not included in the growth model development that the estimates reflect gross accumulation.

Thanks for this suggestion. You are correct that it is carbon held in the aboveground live tree biomass. We have added this clarification to the text. Please note that this information is included in our methods (see line 393):

“Lastly, we only considered live biomass. We did not consider dead wood which is likely removed from many systems via precommercial thinning and was a relatively small proportion of total aboveground carbon stocks.”

Respectfully, we prefer not to add “live” to the title as we have already extended the title (to include aboveground & forests) per the reviewers’ former suggestions, and the title is already longer than the recommended title length at *Nature Communications*. However, we have made sure to add this clarification in the abstract, which is presumably what a reader would read immediately after seeing the title.

Second, while you mentioned repeat measures in your discussion of future assessments and the limitations of the data compiled for use in the analysis, it would be worth expanding on the use of repeat measures as a means to empirically obtaining rates of change in live trees and also tree status, particularly given changes in the drivers of tree growth and mortality.

Thanks for this suggestion. We have added a brief discussion of the importance of repeat measures to the Supporting Information per your recommendation. For your convenience, we have pasted this text below:

Value of repeat measurements

While repeat measurements (i.e., multiple measurements made on the same stand across time) are included in our dataset, they are the minority. That is, most aboveground carbon measurements are taken at single points in time without the stand being revisited in the future. Greater collection and availability of repeat measurements would greatly improve understanding of aboveground carbon accumulation, as additional processes such as mortality can be better accounted for. Further, repeat measurements provide understanding of stand-level patterns of growth, which can improve the ability of analyses such as ours to assess the relative drivers of carbon accumulation. For those empirically estimating standing aboveground carbon in monoculture plantations, we highly recommend the use of

repeat measurements, which will greatly advance our understanding of how biological, environmental, and human factors influence forest growth.

Finally, while the study is global in scope I am still not convinced that there is anything particularly novel about the methods or analysis that warrant consideration in Nature Communications and I would suggest the authors consider a more specialized outlet.

Thanks for this suggestion, but we feel that Nature Communications (open sourced, wide viewership) is an appropriate venue for our article. We'd note that expansion of monoculture plantations as "natural climate solutions" is currently of interest beyond the traditional forestry community, and it's important to us that our conclusions reach a wide-ranging audience.

Reviewer #2 (Remarks to the Author):

REVIEW OF THE MANUSCRIPT 348060_0_art_file_6181187_r55x7g (previously submitted to Nature Climate Change as 23668_0_art_0_qvb1f9)

Rates and drivers of aboveground carbon accumulation in global monoculture plantation forests (in the previous submission it was Rates and drivers of carbon accumulation in global monoculture plantations) by Bukoski et al.

The authors present a study that has the objective to estimate aboveground carbon accumulation in global monoculture plantations. Their objective is interesting of course and the article is original and the results are relevant.

I had already reviewed a previous version of this manuscript for a related journal. The authors considered most of my previous comments/suggestions as well as those from reviewer #1, namely: 1) they are more modest in what concerns objectives and added a good discussion about the limitations of the study (pointed out by the reviewers in relation to the previous submission); 2) they include the data (and also the scripts), therefore we can check them in more detail.

Thank you for noting that we have responded to your comments/suggestions. We appreciated the reviewer's comprehensive feedback before, which we feel has greatly improve the manuscript.

However, I think that there are some issues that need to be better explained:

Data in the csv file should have a description of the variables (some I don't even know what they are and most do not have units) and the references of the papers from which the data were extracted should be provided. I am familiar with some of the data and the above ground carbon values look strange, I would like to check from which publication you got the information. And this may be true for many other readers.

Thank you for this comment. We apologize for not providing the meta-data associated with the dataset, which we have but did not share for the purposes of the review. We are now sharing the final dataset (v

1.0) that will be published, including meta-data that provides descriptions of the variables that have been included. Note that there are additional sheets that link the aboveground carbon “observations” to “sites” information (via “site_id”) and “references” (via “study_id”).

I still have a problem with the interpretation of the asymptote estimates as they are biased by the young ages of the plots measured for some species. But the authors discuss this problem in the discussion section, so it is better than in the previous submission.

Thank you for noting that our discussion of the issue is better than in the previous submission.

We have added several additional clarifications to the text that should further alleviate the issue of low asymptotic values:

- Our asymptote values appear low because they are fixed effects for the population level asymptotic value (i.e., mean AGC across all sites). Examining predicted values on individuals in our dataset yields aboveground carbon values that exceed the population level asymptote (e.g., for Pinus, a maximum predicted value of 164 Mg C / ha, almost double that of the population level asymptote). While our site level random effects are available for our training data, they are not available for locations in which we have no training data and the population level fixed effects are the best estimates in these cases. We have added text to clarify this point and have added the predicted vs. observed figure for Pinus (see below) to the Supporting Information.
- In your prior review, you recommended against truncating the data to the 95th percentile, which could further bias our asymptote estimates. We have thus included all data in our model parameterization per your recommendation.
- Given your comments, we re-examined observations in our database with anomalously low aboveground carbon values at “old” ages. We found a handful of observations that were greater than 20 years of age yet had accumulated < 5 Mg AGC / ha. Backtracking these observations to their associated studies, we found that they were reported in the large compilations by Guo and Ren, and by Schepaschenko and colleagues. The original studies reporting these observations were in either Mandarin or Russian and we could not explain the low AGC values. We have therefore removed these observations from the database. It is worth noting that these observations were really only for Populus and Pinus and did not significantly impact our results. We have also added additional text to the Supplementary Information to describe this procedure (see below).
- We have also added additional text to discourage readers from treating our model parameters as “capital T truth.” That is, all growth function forms will have trade-offs, and our nonlinear modeling should be understood as one attempt at beginning to understand how carbon accumulation varies across these different plantation types. Of course, there are other growth functions (e.g., the Schumacher or Gompertz functions) and statistical approaches, which we hope other researchers will explore given the availability of our dataset.

Additional text added to Supplementary Information:

“For both (the Chinese and Russian) data compilations, we judiciously reviewed observations from young stands with anomalously high aboveground carbon values, as well as old stands with anomalously low aboveground carbon values. When we could not verify an explanation for the anomalously high or low values, we dropped the observations from our database. These anomalously low values were approximately 2% of the total observations in our database and were primarily for Populus plantations in China that had less than 5 Mg AGC/ha after 25 years of growth. Given that Populus is a fast-growing genus and we cannot explain the low aboveground carbon values, we elected to drop them, which risk biasing our growth function parameters.”

Supplementary Figure 6 | Predicted versus observed aboveground carbon for the *Pinus* growth model parameters. Here, random effects estimated for site are incorporated into the predicted aboveground carbon values. Despite unexplained variation in the data, we see the data cluster around the 1-to-1 line (shown in red), and that, for observations in our dataset, maximum aboveground carbon stocks are estimated at ~ 164 Mg AGC ha⁻¹ after incorporating site-level effects.

Lines 122-134 – it is not clear to which statistical test the authors are referring to when they mention the drivers of aboveground carbon accumulation rates, and I was not able to deduce it from the

information provided in the Methods section. Maybe the t-tests for the significance of the parameters in the linear models? Please improve the quality of these paragraphs

Thanks for this comment, which has helped identify a potential shortcoming in our statistics. Before, we were using t-tests for significance of the parameters. However, this is an anti-conservative approach for deriving p-values (risk high Type I errors) in a mixed effects modeling framework.

We have adjusted these statistics to t-tests and their associated p-values using the Satterthwaite method for approximating the denominator degrees of freedom, with all models fitted using REML (as implemented in the {lmerTest} package in R). This approach is less anti-conservative (risk of Type I error does not significantly differ from 0.05), particularly for larger datasets such as ours (thousands of observations). For more information, see:

- Luke, S.G. Evaluating significance in linear mixed effects models in R. *Behavior Research Methods* 49(4): 1494-1502.

Generally, our results have not changed. Nevertheless, we have updated the statistics, text, and tables throughout the manuscript to reflect these adjustments.

Again, thank you for this comment, which we feel has strengthened the statistical analysis of our study.

Line 147 – it is really strange that planting density does not have an effect, I think the problem must be that there are several multicollinearities among the different drivers and you tested one driver at a time added to a model with age. The impact of planting density may be masked by other driver(s). I “played” a bit with the data and fixing the species and the ages, we can see some increasing tendency with planting density. Therefore, fitting linear mixed models of aboveground c accumulation as a function of just age and one driver may not be a good option. Please discuss this detail.

Thanks for this suggestion. We have added additional discussion of this detail in the Supporting Material. We agree that by fixing species and age, we would expect aboveground carbon to increase with density. However, as we mentioned before, it will also depend on the mean tree diameter in the stand (density-size inverse relationship). Fully explaining this result would require stand-level mean tree diameter, which we were unable to collect consistently from the studies. We have added the text below to the Supporting Information.

The effect of planting density on aboveground carbon

An unexpected result that warrants additional discussion is the non-significance of “planting density” in our Full Model 2 (see the “Drivers of aboveground carbon accumulation rates” section). This result can potentially be explained by several factors. First, it is worth reiterating that this driver was planting density (i.e., at $t = 0$) rather than actual stand density at the time of field inventory, and there may be minor differences between the two due to either density-dependent mortality (i.e., self-thinning) or management-driven thinning. Second, most plantations in our dataset were managed for production of harvested wood products. It is therefore likely that planting spacing was optimized to maximally capture growing space across studies, and understocked plantations were likely poorly represented in our data. Lastly, fully stocked plantations are known to exhibit strong size-density tradeoffs (which underpins well-established silvicultural tools such as Reineke’s Stand Density Index). Assuming that plantations have been planted to maximally occupy growing space, it is likely that the

effect of low-density plantings on aboveground carbon are partially offset by greater mean stem diameters at the stand-level. Fully teasing out this relationship would require data on mean stand diameter, which we were unable to collect consistently across studies. Nevertheless, exploratory analyses of the data suggest that when fixing both genus and age (e.g., filtering to all *Pinus* plantations of 20 years of age), we see some tendency towards greater aboveground carbon with increasing planting density.

Line 500 – I suppose it is better to say “... functional types with more than 100 observations...” than “... functional types with greater than 100 observations...”

Thanks for this suggestion. We have adjusted the text as recommended.

Lines 522-528 – about fixing the b and m parameters of the Richards function: I agree with fixing $b=1$ as it just indicates that $abc=0$ when $t=0$, which is fine, but the m parameter (I would not call it an allometric constant, it is a shape parameter) is related to the shape of the curve, namely with the location of the inflection point, I don't see a reason to fix it. You can go to <http://home.isa.utl.pt/~joaopalma/modelos/fgfp/> and check the important role of m in the shape of the curve (and, to a certain, extent, in the growth rate).

Thanks for this suggestion. We have several reasons for fixing the m parameter:

- First, it is worth noting our understanding of the “ m ” parameter, just to be sure we are on the same page. The addition of the m parameter was a generalization of the von Bertalanffy growth function that provides the Chapman-Richards growth function more flexibility and better fits to empirical data (Richards 1959). In the Bertalanffy function, m is fixed at $2/3$, effectively setting the exponent of the C-R function at 3. The m value of $2/3$ is based on assumptions about the rates of anabolism and catabolism, which has been substantiated by empirical evidence. The addition of the m parameter is viewed as beneficial for flexible fits to data but detrimental for “explaining” growth patterns. As noted in Zeide 1993: “... Rolfe A. Leary remarked that compared with Bertalanffy's equation, the one by Chapman-Richards is ‘a giant leap backwards from explanation to description’.” Given our primary motivation of *explaining* variation in patterns of carbon accumulation across plantation types, it makes most sense to fix the m parameter.
- The m and k parameters are also closely related (see Burkhardt and Tome, page 121). By fixing one, we force more of the growth rate information of the curve to be carried by k , allowing us to compare k more safely across models (i.e., one cannot compare k for different values of m). Allowing both m and k to vary would greatly complicate interpretation of our results. [Note that we have confirmed this with a biometrician at Yale University, who was also a former regional president of the International Biometric Society].
- We explored the possibility of allowing m to vary for a subset of our models, and m was driven to ~ 0 , which in effect turns the C-R function into the monomolecular growth function. The monomolecular growth function does not have an inflection point (i.e., it is not sigmoidal), which does not align with established understanding of biological growth. While this m might produce a better “fit” to the data, we strongly prefer to retain a theoretical basis in our models.

- We foresee significant convergence issues (which are known to exist with the Chapman-Richards model; see Zeide 1993) when allowing m to vary, as well as very large time investments in identifying appropriate starting values across the three parameters (A , m , & k).

For these four reasons, we strongly prefer fixing m . However, we have made several adjustments based on this comment:

1. We have re-estimated the parameters while fixing m at 0.67 rather than 0.5, which produces the Bertalanffy “special case” of the Chapman-Richards model. We now feel that fixing m at 0.67 (as opposed to $m = 0.5$) is better substantiated by the literature.
2. We have added a new figure that better summarizes yield (Mg AGC / ha) and growth rates (dy/dt) as a function of age (y), as well as growth rate as a function of stand-level aboveground carbon ($y(t)$) for each of the fourteen plantation types (pasted below for convenience). This figure supplements the comparison of the A and k parameters across plantation types.
3. We agree that it’s important to explore how alternative values of m might impact our results. Thus, we have added a sensitivity analysis to our Supplementary Information. Rather than fixing m at a single value (0.67), we re-estimated the A and k parameters using each of three values for m (0.5, 0.67, and 0.75), which correspond to exponents of 2, 3, & 4 in the Chapman-Richards equation, respectively. We identify the “best fit” parameterization using the lowest RMSE. Note that while in many cases $m = 0.5$ is the “best fit” parameterization, we have retained m at 0.67 in our manuscript given the empirical evidence for this as shown by von Bertalanffy.
4. We have added discussion and a table to the Supporting Information that reports these “best fit” parameters. We like this approach for two reasons: i) it helps show how the mathematical structure of the model effects the results, and ii) it pulls readers away from a strict biological interpretation of our model parameters (i.e., A is not an undeniable “True” asymptotic growth limit).
5. Lastly, we have added additional text clarifying our above justifications and all changes to our methods.

We hope that this is an acceptable compromise that addresses the reviewer’s concerns.

Response to reviewer comments, second round –

Reviewer #2 (Remarks to the Author):

I carefully analysed the revised version of the manuscript and concluded that the authors have incorporated all the suggestions made.